# GRADIENT CONSTRAINED SHARPNESS-AWARE PROMPT LEARNING FOR VISION-LANGUAGE MODELS

## ABSTRACT

This paper targets a novel trade-off problem in generalizable prompt learning for vision-language models (VLM), *i.e.*, improving the performance on unseen classes while maintaining the performance on seen classes. Comparing with existing generalizable methods that neglect the seen classes degradation, the setting of this problem is stricter and fits more closely with practical applications. To solve this problem, we start from the optimization perspective, and leverage the relationship between loss landscape geometry and model generalization ability. By analyzing the loss landscapes of the state-of-the-art method and vanilla Sharpness-aware Minimization (SAM) based method, we conclude that the trade-off performance correlates to both **loss value** and **loss sharpness**, while each of them is indispensable. However, we find the optimizing gradient of existing methods cannot maintain high relevance to both loss value and loss sharpness during optimization, which severely affects their trade-off performance. To this end, we propose a novel SAM-based method for prompt learning, denoted as Gradient Constrained Sharpness-aware Context Optimization (GC-SCoOp), to dynamically constrain the optimizing gradient, thus achieving above two-fold optimization objective simultaneously. Extensive experiments verify the effectiveness of GCSCoOp in the trade-off problem. Code is available at https://anonymous.4open.science/r/GCSCoOp-1323.

## 1 INTRODUCTION

Prompt learning comes to the fore as a parameter-efficient fine-tuning (PEFT) method to adapt pre-trained vision-language models (VLM) (Radford et al., 2021; Jia et al., 2021) to downstream visual tasks. Conventional prompt learning, *e.g.*, Context Optimization (CoOp) (Zhou et al., 2022b), often overfits on the training data and severely undermines VLM's original generalization ability. Since this problem leads to a significant accuracy decline on unseen classes even within the same downstream task, it obviously limits the applicability of the learned prompt in various real scenarios.

To this end, recent approaches (Zhou et al., 2022a; Zhu et al., 2022; Yao et al., 2023) target to learn generalizable prompts. However, comparing with original CoOp, existing generalizable prompt learning approaches suffer from significant degradations of the discriminative ability on seen (trained) classes in downstream tasks. This phenomenon has also been acknowledged by the state-of-the-art KgCoOp (Yao et al., 2023) as a critical limitation of it. The primary purpose of prompt learning is to improve VLM's adaptation ability on designated downstream tasks, especially on the trained data. Therefore, solely pursuing the improvement on unseen classes without considering the degradation on seen classes is undesirable.

Based on above discussions, we redefine a **trade-off problem** for generalizable prompt learning, *i.e.*, improving the performance on unseen classes while maintaining the performance on seen classes. This problem setting has a tighter restriction and fits more closely with real application scenarios. In this paper, we tend to explore the trade-off problem from the optimization perspective. Extensive works (Keskar et al., 2016; Dziugaite & Roy, 2017; Jiang et al., 2019) have demonstrated the relationship between loss landscape geometry and model generalization ability, and indicated that parameters located in flatter minima were usually more generalizable. For this reason, we first observe the loss landscape of vanilla CoOp and the state-of-the-art KgCoOp. As shown in Fig.1, we find the seen and unseen classes performance of the same downstream task are tightly associated

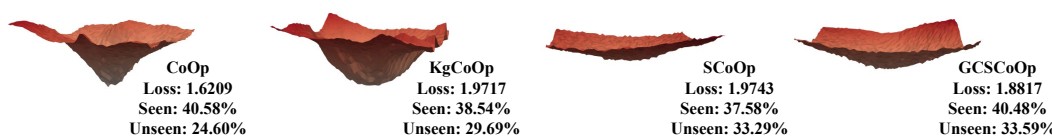

Figure 1: Loss landscapes and performances of the prompts learned by CoOp, KgCoOp, SCoOp and GCSCoOp on FGVCAircraft dataset, respectively. We plot the results by following the instruction in Li et al. (2018). In order to display the plots more aesthetically, we align them horizontally rather than in the same coordinate system. We explicitly give the loss value of the minima as the reference.

with the loss value and loss sharpness, respectively. Comparing with CoOp, KgCoOp indeed provides a flatter loss curve but the converged loss value is much higher, which indicates that KgCoOp weakens the original discriminative ability and leads to the seen classes degradation. This observation reveals an intuitive solution to the trade-off problem: **only if loss value and loss sharpness can be optimized appropriately and simultaneously, the learned prompt can achieve satisfying trade-off**.

To explicitly optimize loss sharpness, we first leverage the widely-used Sharpness-aware Minimization (SAM) (Foret et al., 2021; Wu et al., 2020) in prompt learning, and propose *Sharpness-aware Context Optimization (SCoOp)*. SAM optimizes loss sharpness by penalizing the neighborhood of minima to have uniform low loss, and has achieved great success in training overparameterized models (Arora et al., 2019; Belkin, 2021; Vicol et al., 2022). Nevertheless, we find that vanilla SAM still fails to improve the generalization trade-off in prompt learning. Specifically, we compute the cosine distance between the gradient directions of the loss ($\mathcal{L}_{SAM}$) in SAM and the loss ($\mathcal{L}_{ERM}$) in Empirical Risk Minimization (ERM), where ERM optimizes parameters only according to loss values. We observe that sometimes, especially at the early stage of training, **the directions of $\nabla \mathcal{L}_{SAM}$ and $\nabla \mathcal{L}_{ERM}$ have very limited consistency (a large direction angle), or even conflict (the angle is larger than** $90°$). This phenomenon demonstrates that SAM cannot always optimize loss value and loss sharpness simultaneously in prompt learning task.

Different from overparameterized models that theoretically have multiple optimal solutions, learnable prompts have very few tunable parameters, *i.e.*, a set of token vectors in text inputs, which means the optimal solution in prompt learning task is very limited. Therefore, the loss value is much more sensitive with regard to the direction of optimizing gradient in prompt learning. And then, the above inconsistency problem will lead to a much severe affect in prompt learning than in overparameterized model training. Directly optimizing prompts through $\nabla \mathcal{L}_{SAM}$ will cause the prompt parameters fail to converge to a sufficient low loss value, and consequently leads to a significant decrease on the performance of seen classes. (We provide detailed empirical study and in-depth analysis in Section 4.4 and Appendix A.1).

Based on above analysis, we design Gradient Constrained Sharpness-aware Minimization (GC-SAM) and propose the *Gradient Constrained Sharpness-aware Context Optimization (GCSCoOp)* for prompt learning. Based on the cosine similarity between the direction of $\nabla \mathcal{L}_{SAM}$ and $\nabla \mathcal{L}_{ERM}$, GCSAM sets appropriate thresholds to distinguish the optimization objective under different conditions. For each condition, GCSAM constrains the optimizing gradient to have high relevance to both loss value and loss sharpness based on the corresponding objective, *i.e.*, the direction of the gradient has high consistency with both optimizing directions *w.r.t.* loss value and loss sharpness. As shown in Fig.1, comparing with SCoOp, GCSCoOp converges to a lower loss value while penalizing the loss sharpness simultaneously. Extensive experiments demonstrate that GCSCoOp achieves better generalization trade-off performance comparing with state-of-the-art methods and SCoOp.

## 2 RELATED WORKS

**Pre-trained vision-language models (VLM).** Recent works have proposed multiple VLMs, such as CLIP (Radford et al., 2021), ALIGN (Jia et al., 2021), FILIP (Yao et al., 2021) and LiT (Zhai et al., 2022), to excavate semantic relationship between visual and language modalities through large-scale image-text pairs pre-training. Such model usually consists of an image-encoder (*e.g.*, ViT (Dosovitskiy et al., 2021) or ResNet (He et al., 2016)) and a text-encoder (*e.g.*, Transformer

(Vaswani et al., 2017)). Benefited from the rich nature language supervision, VLM shows great potential in open-world visual understanding. We adopts CLIP as the foundation VLM for research.

**Prompt learning.** Prompt learning is an emerging parameter-efficient fine-tuning (PEFT) technique for downstream task adaptation. Inspired by NLP-related researches, VLM prompt learning exploits extra text prompt information to obtain more comprehensive and task-related text inputs, thereby improving VLM's adaptation performance on downstream tasks by fine-tuning very few parameters. Zero-shot CLIP (Radford et al., 2021) initially utilized hand-crafted prompt to embellish original class names (*e.g.*, "a photo of a [class]") and achieved zero-shot image classification. CoOp (Zhou et al., 2022b) was the earliest work that exploited a set of tunable token vectors to learn task-specific prompt through few-shot learning. To address the poor generalization problem in CoOp, subsequent works started to learn generalizable prompts by proposing instance-level prompts (CoCoOp (Zhou et al., 2022a)) or adopting manual prompts as general knowledge guidance (ProGrad (Zhu et al., 2022) and KgCoOp (Yao et al., 2023)). Besides, there were also some works achieved better prompt learning from other different perspectives, e.g., multi-modal prompts (Khattak et al., 2023; Roy & Etemad, 2023; Xu et al., 2023) or multiple prompts that provided comprehensive description (Lu et al., 2022; Chen et al., 2023; Bulat & Tzimiropoulos, 2023; Ge et al., 2023). This work mainly focuses on the basic single text-based prompt learning for research.

**Sharpness-aware Minimization (SAM).** The optimization strategy of SAM (Foret et al., 2021; Wu et al., 2020; Kwon et al., 2021; Zhuang et al., 2022) was initially proposed to improve the generalization ability of overparameterized models. Since this kind of models can easily fall into the local minima during training phase, SAM leveraged the relationship between the geometry of loss landscape and the model generalization ability (Keskar et al., 2016; Dziugaite & Roy, 2017; Jiang et al., 2019), and sought the minima that has flatter loss curve. PFLAT (Shen et al., 2023) claimed that large language model (LLM) with flatter loss landscape can lead to robust prompt selection in LLM. One of our concurrent work BSAPT (Fan et al., 2023) also directly leveraged SAM in visual prompts for cross-domain few-shot learning. Different from BSAPT, we provide comprehensive analysis for applying SAM in prompt learning, and propose an improved SAM (GCSAM) to solve the newly prompt generalization trade-off problem, which associates more tightly with real applications.

## 3 METHODOLOGY

This work proposes Gradient Constrained Sharpness-aware Context Optimization (GCSCoOp) to learn generalizable prompts for VLM. In this section, we first give the preliminary concepts of the used vision-language model CLIP and the baseline approach Context Optimization (CoOp) (Sec.3.1). Then, we provide the detailed implementation of Sharpness-aware Minimization (SAM) and the corresponding Sharpness-aware Context Optimization (SCoOp) for prompt learning (Sec.3.2). Finally, we introduce the designed Gradient Constrained Sharpness-aware Minimization (GCSAM) and the overall construction of GCSCoOp (Sec.3.3).

### 3.1 PRELIMINARIES

**Contrastive language-image pre-training (CLIP).** CLIP consists of an image-encoder $\mathcal{I}$ and a text encoder $\mathcal{T}$. By training with 400 million image-text training pairs, CLIP aims to align the image and text features extracted by $\mathcal{I}$ and $\mathcal{T}$ via contrastive learning. In this paper, we denote $\boldsymbol{x}$ and $\boldsymbol{f}$ as image inputs and the corresponding image features, while $\boldsymbol{t}$ and $\boldsymbol{w}$ as text inputs and text features, respectively. The pre-trained CLIP can be easily adapted to zero-shot image classification task. Specifically, the text inputs $\boldsymbol{t}$ are described by combining class name with hand-crafted prompts such as "a photo of a [class]". With $M$ categories in the task, there are a set of text features $\boldsymbol{w} = \{\boldsymbol{w}_j\}_{j=1}^M$ generated by $\mathcal{T}$ (one class name corresponds to one text input). Given an image $\boldsymbol{x}$, visual encoder extracts the image feature as: $\boldsymbol{f} = \mathcal{I}(\boldsymbol{x})$. Then, the prediction probability of the $m$-th class can be derived as:

$$p(m|\boldsymbol{x}) = \frac{\exp(\cos(\boldsymbol{f}, \boldsymbol{w}_m)/\tau)}{\sum_{j=1}^M \exp(\cos(\boldsymbol{f}, \boldsymbol{w}_j)/\tau)}, \tag{1}$$

where $\cos(\cdot, \cdot)$ indicates the cosine similarity and $\tau$ is a temperature parameter learned by CLIP.

**Context Optimization (CoOp).** CoOp further enhances VLM's adaptation ability by exploiting a set of tunable token vectors $\boldsymbol{v} = \{\boldsymbol{v}_i\}_{i=1}^N$ to learn task-specific prompt through few-shot learning

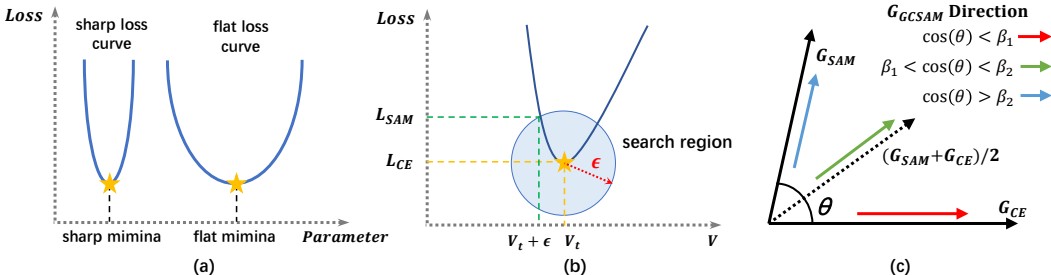

Figure 2: Schematics of the motivation of SCoOp and GCSCoOp.

on downstream tasks, where $N$ indicates the token vector length. Therefore, the text inputs of $m$-th class can be interpreted as $\boldsymbol{t}_m = \{\boldsymbol{v}_1, \boldsymbol{v}_2, ..., \boldsymbol{v}_N, \boldsymbol{c}_m\}$, where $\boldsymbol{c}_m$ is the $m$-th class name. CoOp conducts prompt learning with Empirical Risk Minimization (ERM). Specifically, given a few-shot sample $\boldsymbol{x}$ and ground-truth label $y_m$, CoOp optimizes prompt parameters $\boldsymbol{v}$ via cross-entropy loss $\mathcal{L}_{CE}$ between the prediction probability and the label (all of the CLIP parameters are fixed):

$$\mathcal{L}_{CE}(\boldsymbol{v}) = -\sum_m y_m \log p(m|\boldsymbol{x}), \quad p(m|\boldsymbol{x}) = \frac{\exp(\cos(\mathcal{I}(\boldsymbol{x}), \mathcal{T}(\boldsymbol{t}_m))/\tau)}{\sum_{j=1}^M \exp(\cos(\mathcal{I}(\boldsymbol{x}), \mathcal{T}(\boldsymbol{t}_j))/\tau)}. \quad (2)$$

### 3.2 SHARPNESS-AWARE CONTEXT OPTIMIZATION

Empirical study has demonstrated that the learned prompt of CoOp tends to overfit on the source (seen) training data, thus undermining the generalization ability of VLM on target (unseen) data. Existing generalizable approaches indeed improve the performance of the learned prompt on unseen classes, but they neglect the seen classes degradation. To associate more tightly with practical applications, we redefine a new generalization trade-off problem for prompt learning, *i.e.*, improving the performance on unseen classes while maintaining the performance on seen classes.

Many prior works (Keskar et al., 2016; Dziugaite & Roy, 2017; Jiang et al., 2019; Foret et al., 2021) have indicated that overparameterized model with flatter loss curve (as illustrated in Fig.2 (a)) usually has better generalization performance. Therefore, we explore the trade-off problem from the optimization perspective by leveraging the relationship between loss landscape geometry and model generalization ability. According to the previous discussions, we analyze the failure cases of CoOp and state-of-the-art KgCoOp, and conclude that seen and unseen classes performance highly related to the value and sharpness of the loss landscape, respectively. Specifically, vanilla CoOp optimizes the prompt parameters $\boldsymbol{v}$ directly through the gradient $G_{CE} = \frac{\partial \mathcal{L}_{CE}}{\partial \boldsymbol{v}}$, and this approach only targets to minimize the loss value, which leads to a sharp loss curve during the optimization procedure. KgCoOp achieves a flatter loss curve but the converged loss value is much higher than CoOp, thus undermining the discriminative ability of learned prompt on seen classes. This observation inspires us to optimizing both loss sharpness and loss value simultaneously.

We then leverage Sharpness-aware Minimization (SAM) in prompt learning, which denoted as *Sharpness-aware Context Optimization (SCoOp)*. Fig.2 (a) reveals a key characteristic of flat loss curve: the neighboring parameters of the minima have uniform low loss. Based on this observation, SCoOp can explicitly optimize loss sharpness by penalizing the largest loss value in the neighborhood of minima to induce the prompt parameters converged at a flat minima. As illustrated in Fig.2 (b), given the current prompt parameters $\boldsymbol{v}^k$, we seek the largest loss value $\mathcal{L}_{SAM}$ in the minima-centered region by perturbing $\boldsymbol{v}^k$ to $\boldsymbol{v}^k + \boldsymbol{\epsilon}$, where the magnitude of $\boldsymbol{\epsilon}$ is controlled by the *perturbation radius $\rho$*. By updating $\boldsymbol{v}^k \rightarrow \boldsymbol{v}^{k+1}$ through the gradient $G_{SAM} = \frac{\partial \mathcal{L}_{SAM}}{\partial \boldsymbol{v}^k}$, we can finally obtain a flatter loss curve at the converged minima.

Specifically, the implementation of SCoOp involves a min-max optimization procedure, which can be described as:

$$\min_{\boldsymbol{v}} \mathcal{L}_{SAM}(\boldsymbol{v}) \quad \text{where} \quad \mathcal{L}_{SAM}(\boldsymbol{v}) \triangleq \max_{||\boldsymbol{\epsilon}||_p \leq \rho} \mathcal{L}_{CE}(\boldsymbol{v} + \boldsymbol{\epsilon}), \quad (3)$$

where $\rho \geq 0$ is the hyperparameter of perturbation radius and $p$ indicates the $p$-norm (typically $p = 2$). Since $\mathcal{L}_{SAM}$ is induced by the maximum loss of $\mathcal{L}_{CE}$, the key problem to solve Eq.3 is to

find the perturbation $\epsilon$. With a relative small $\rho$, the inner maximization problem can be approximated by first order Taylor expansion and the optimal perturbation can be derived as:

$$\hat{\epsilon} = \underset{||\epsilon|| \leq \rho}{\arg\max} \, \mathcal{L}_{CE}(\boldsymbol{v} + \epsilon) \approx \underset{||\epsilon|| \leq \rho}{\arg\max} \, \mathcal{L}_{CE}(\boldsymbol{v}) + \epsilon^{\top} \nabla \mathcal{L}_{CE}(\boldsymbol{v}) = \rho \frac{\nabla \mathcal{L}_{CE}(\boldsymbol{v})}{\|\nabla \mathcal{L}_{CE}(\boldsymbol{v})\|}. \quad (4)$$

With the optimal perturbation $\hat{\epsilon}$, we can obtain the maximum loss value in the neighborhood of $\boldsymbol{v}$:

$$\mathcal{L}_{SAM}(\boldsymbol{v}) = \mathcal{L}_{CE}(\boldsymbol{v} + \rho \frac{\nabla \mathcal{L}_{CE}(\boldsymbol{v})}{\|\nabla \mathcal{L}_{CE}(\boldsymbol{v})\|}). \quad (5)$$

Eq.5 can be easily solved according to Eq.2. Finally, the prompt parameters are straightforwardly optimized by minimizing $\mathcal{L}_{SAM}$ as all the above computation procedures are differentiable.

## 3.3 GRADIENT CONSTRAINED SHARPNESS-AWARE CONTEXT OPTIMIZATION

However, although SCoOp can further improve the generalization ability of the learned prompt by explicitly optimizing loss sharpness, we find this approach still fails on the trade-off problem. Based on previous discussions, we find vanilla SAM easily causes the prompt parameters converge at an insufficient low loss value due to the gradient inconsistency problem, and thus degrading the discriminative ability on seen classes. To guarantee loss sharpness and loss value can be simultaneously penalized during the whole training process, we propose the *Gradient Constrained Sharpness-aware Minimization (GCSAM)*. GCSAM sets thresholds lower bound $\beta_1$ and upper bound $\beta_2$ to distinguish different optimization conditions. For each condition, GCSAM constrains the optimizing gradient to have high enough relevance to the two-fold optimization objective simultaneously.

As illustrated in Fig.2 (c), we compute the cosine similarity of $G_{SAM}$ and $G_{CE}$:

$$cos(\theta) = \frac{G_{SAM} \cdot G_{CE}}{\|G_{SAM}\| \cdot \|G_{CE}\|}. \quad (6)$$

If the direction of $G_{SAM}$ and $G_{CE}$ are highly consistent ($cos(\theta) \geq \beta_2$), GCSAM just keeps the original $G_{SAM}$. If $cos(\theta) \leq \beta_1$, which means $G_{SAM}$ and $G_{CE}$ are almost irrelevant or even conflict. This condition usually occurs at the early stage of training (shown in Fig.3) and the parameters have not converged at minima yet, which indicates that the loss sharpness is unnecessary to be considered at this stage. Thus, GCSAM optimizes $\boldsymbol{v}$ through $G_{CE}$. Otherwise ($\beta_1 < cos(\theta) < \beta_2$), both loss sharpness and loss value should be considered. We constrain $G_{GCSAM}$ by projecting $G_{SAM}$ to the middle direction of $G_{SAM}$ and $G_{CE}$. The overall $G_{GCSAM}$ can be described as:

$$G_{GCSAM} = \begin{cases} G_{CE}, & cos(\theta) \leq \beta_1 \\ \frac{G_{SAM} \cdot G_{\text{mid}}}{\|G_{\text{mid}}\|^2} G_{\text{mid}}, & \beta_1 < cos(\theta) < \beta_2 \\ G_{SAM}, & cos(\theta) \geq \beta_2 \end{cases} \quad (7)$$

where $G_{\text{mid}} = (G_{SAM} + G_{CE})/2$. We then introduce GCSAM as the optimization strategy for prompt learning, denoted as *Gradient Constrained Sharpness-aware Context Optimization (GC-SCoOp)*. GCSCoOp simultaneously takes the two-fold optimization objective into consideration during prompt learning, thus achieving better generalization trade-off performance. Notably, since SCoOp originally involves the computation of $G_{CE}$, GCSCoOp does not need extra gradient computation costs. The pseudo codes of SCoOp and GCSCoOp are provided in Appendix A.2.

## 4 EXPERIMENTS

In this section, we first introduce the datasets, baseline and implementation details used in this paper (Sec.4.1). Then, we conduct the experiments of seen-to-unseen class generalization (Sec.4.2) and cross-datasets generalization (Sec.4.3). Finally, we provide the visualization results of training process and loss landscape (Sec.4.4) and more detailed hyperparameter analysis (Sec.4.5).

## 4.1 DATASETS AND IMPLEMENTATION DETAILS

This work adopts 15 public available image classification datasets as downstream tasks for experiments: ImageNet (Deng et al., 2009), Caltech101 (Fei-Fei et al., 2004), OxfordPets (Parkhi et al.,

Table 1: Performance comparison of Zero-shot CLIP, CoOp, CoCoOp, ProGrad, KgCoOp with the proposed SCoOp and GCSCoOp in seen-to-unseen generalization over 11 datasets. Here *HM* denotes the harmonic mean between seen and unseen accuracy. SCoOp† indicates the best performance SCoOp can achieve by searching the most suitable perturbation radius $\rho$ for each dataset.

| | ImageNet | | | Caltech101 | | | OxfordPets | | |
|---|---|---|---|---|---|---|---|---|---|
| | Seen | Unseen | HM | Seen | Unseen | HM | Seen | Unseen | HM |
| CLIP | 72.42 | 68.13 | 70.21 | 97.29 | 94.10 | 95.67 | 89.47 | 96.81 | 93.00 |
| CoOp | 76.34 | 65.17 | 70.31 | 98.15 | 93.23 | 95.63 | 94.19 | 96.01 | 95.09 |
| CoCoOp | 76.09 | 69.73 | 72.77 | 97.92 | 91.78 | 94.75 | 93.48 | 95.77 | 94.61 |
| ProGrad | **76.72** | 67.80 | 71.98 | **98.30** | 93.96 | 96.08 | 94.59 | 96.98 | 95.77 |
| KgCoOp | 75.89 | 69.55 | 72.58 | 97.72 | 94.32 | 95.99 | 95.16 | 96.61 | 95.88 |
| SCoOp | 76.29 | 71.19 | 73.65 | 97.98 | 93.05 | 95.45 | 95.39 | 97.91 | 96.63 |
| SCoOp† | 76.29 | 71.19 | 73.65 | 97.65 | 95.09 | 96.35 | **95.48** | **98.10** | **96.77** |
| GCSCoOp | 76.42 | **71.28** | **73.76** | 98.11 | **95.27** | **96.67** | 95.43 | 97.97 | 96.68 |

| | StanfordCars | | | Flowers102 | | | Food101 | | |
|---|---|---|---|---|---|---|---|---|---|
| | Seen | Unseen | HM | Seen | Unseen | HM | Seen | Unseen | HM |
| CLIP | 63.87 | **74.97** | 68.98 | 69.23 | **76.74** | 72.79 | 89.42 | 90.68 | 90.05 |
| CoOp | 77.58 | 65.02 | 70.75 | **97.79** | 63.95 | 77.33 | 88.68 | 85.31 | 86.96 |
| CoCoOp | **77.65** | 63.00 | 69.56 | 97.44 | 60.97 | 75.01 | 88.11 | 83.92 | 85.96 |
| ProGrad | 77.11 | 69.89 | 73.32 | 96.52 | 69.86 | 81.05 | 90.11 | 89.56 | 89.83 |
| KgCoOp | 74.16 | 74.64 | 74.40 | 95.85 | 73.19 | 83.00 | 90.53 | 91.01 | 90.77 |
| SCoOp | 71.31 | 74.85 | 73.04 | 97.06 | 72.58 | 83.05 | 90.92 | 92.07 | 91.49 |
| SCoOp† | 76.14 | 71.91 | 73.96 | 97.06 | 72.58 | 83.05 | 90.91 | **92.15** | **91.53** |
| GCSCoOp | 75.43 | 74.06 | **74.74** | 97.44 | 72.93 | **83.42** | **90.96** | 92.07 | 91.51 |

| | FGVCAircraft | | | SUN397 | | | DTD | | |
|---|---|---|---|---|---|---|---|---|---|
| | Seen | Unseen | HM | Seen | Unseen | HM | Seen | Unseen | HM |
| CLIP | 27.55 | 33.29 | 30.15 | 69.40 | 75.56 | 72.35 | 53.36 | 51.69 | 52.51 |
| CoOp | 39.94 | 24.62 | 30.46 | 80.58 | 63.90 | 71.28 | 79.82 | 45.17 | 57.69 |
| CoCoOp | **41.80** | 25.08 | 31.35 | 79.38 | 67.99 | 73.24 | 79.94 | 42.55 | 55.54 |
| ProGrad | 40.30 | 25.81 | 31.47 | 81.11 | 71.31 | 75.89 | 76.85 | 51.89 | 61.95 |
| KgCoOp | 38.72 | 29.63 | 33.57 | 80.71 | 76.28 | 78.43 | **80.44** | 56.69 | **66.51** |
| SCoOp | 37.70 | **33.69** | **35.58** | 81.09 | 77.22 | 79.11 | 68.87 | 56.52 | 62.09 |
| SCoOp† | 37.70 | **33.69** | **35.58** | 80.91 | **78.04** | **79.45** | 77.58 | **57.09** | 65.78 |
| GCSCoOp | 39.28 | 32.45 | 35.54 | **81.17** | 77.15 | 79.11 | 80.32 | 55.39 | 65.57 |

| | EuroSAT | | | UCF101 | | | **AVG** | | |
|---|---|---|---|---|---|---|---|---|---|
| | Seen | Unseen | HM | Seen | Unseen | HM | Seen | Unseen | HM |
| CLIP | 50.19 | 69.90 | 58.43 | 68.10 | 75.12 | 71.44 | 68.21 | 73.36 | 70.51 |
| CoOp | 91.25 | 47.26 | 62.27 | **85.20** | 56.05 | 67.62 | **82.68** | 64.15 | 71.40 |
| CoCoOp | **92.14** | 51.33 | 65.93 | 84.25 | 59.17 | 69.52 | 82.56 | 64.66 | 71.66 |
| ProGrad | 91.09 | 56.21 | 69.52 | 84.59 | 67.03 | 74.79 | 82.48 | 69.12 | 74.70 |
| KgCoOp | 88.15 | 60.42 | 71.70 | 84.00 | **75.37** | 79.45 | 81.94 | 72.52 | 76.57 |
| SCoOp | 71.46 | 58.03 | 64.05 | 84.09 | 70.63 | 76.77 | 79.29 | 72.52 | 75.54 |
| SCoOp† | 85.09 | 66.26 | 74.50 | 85.06 | 71.95 | 77.96 | 81.81 | 73.46 | 77.14 |
| GCSCoOp | 87.91 | **69.92** | **77.89** | 84.56 | 75.01 | **79.50** | 82.46 | **73.95** | **77.67** |

2012), StanfordCars (Krause et al., 2013), Flowers102 (Nilsback & Zisserman, 2008), Food101 (Bossard et al., 2014), FGVCAircraft (Maji et al., 2013), SUN397 (Xiao et al., 2010), DTD (Cimpoi et al., 2014), EuroSAT (Helber et al., 2019), UCF101 (Soomro et al., 2012), ImageNetV2 (Recht et al., 2019), ImageNet-Sketch (Wang et al., 2019), ImageNet-A (Hendrycks et al., 2021b) and ImageNet-R (Hendrycks et al., 2021a). Notably, the latter 4 datasets are only applied as the target domain for cross-domain generalization. These datasets constitute a comprehensive benchmark, which including classification on generic objects, scenes, actions, satellites, textures and fine-grained categories. The comprehensive dataset statistics are demonstrated in Appendix A.3.1.

The implementation of SCoOp and GCSCoOp are based on CoOp. Specifically, we utilize CLIP as the foundation model with visual backbone ViT-B/16. The learnable token vector length is set to 4 with the initialization template "a photo of a [class name]". The training phase also referred to

Table 2: Performance comparison of VLM prompt learning methods under the cross domain transfer learning setting. Here, V2, Sketch, A and R indicate the ImageNetV2, ImageNet-Sketch, ImageNet-A and ImageNet-R datasets, respectively.

| | Source | Target | | | | |
|---|---|---|---|---|---|---|
| | ImageNet | V2 | Sketch | A | R | **AVG** |
| CoOp | 71.5 | 64.44 | 47.61 | 49.53 | 74.98 | 59.14 |
| CoCoOp | 71.51 | 64.38 | 48.3 | 50.26 | 75.57 | 59.63 |
| ProGrad | **71.72** | 64.27 | 48.1 | 49.72 | 75.84 | 59.48 |
| KgCoOp | 70.76 | 63.66 | 48.69 | 50.37 | 76.76 | 59.87 |
| SCoOp | 71.2 | 64.18 | 49.06 | 51.15 | 76.87 | 60.32 |
| GCSCoOp | 71.31 | **64.51** | **49.26** | **51.35** | **77.11** | **60.56** |

few-shot learning and all the experiments are conducted with 16-shots as default unless mentioned. We train GCSCoOp for 200 epochs with 128 batch size. GCSCoOp includes three important hyperparameters. Specifically, we set the perturbation radius $\rho = 0.1$, the lower bound $\beta_1 = 0.5$ and upper bound $\beta_2 = 0.8$ as default. We find this setting can fit with most of the datasets. For other datasets, we just simply adjust the hyperparameters according to their characteristics. We provide the detailed discussion and hyperparameters of each dataset in the Appendix A.3.2. All the other training details (*e.g.*, learning rate, momentum and weight decay) are the same with CoOp.

We compare GCSCoOp with five related methods: Zero-shot CLIP, CoOp, CoCoOp, ProGrad and KgCoOp, and also SCoOp that optimized with vanilla SAM. Notably, ProGrad and KgCoOp are two newly and most-related approaches that also focuses on learning generalizable prompts under vanilla single text-based prompt learning framework. All the methods are reproduced under the same experimental settings (except only 20 epochs for training CoCoOp on ImageNet since the huge computation duration) and the final results are averaged over three random seeds for fair comparison.

## 4.2 SEEN-TO-UNSEEN CLASS GENERALIZATION

Similar to previous works, we split each dataset into disjoint seen and unseen classes. To verify the generalization ability, prompts are only learned with seen classes and evaluated on both seen and unseen classes. Table.1 demonstrates the seen and unseen classes performances of all compared methods with SCoOp and GCSCoOp among 11 datasets. As reported in the averaged performance, the proposed SAM-based methods (SCoOp and GCSCoOp) largely improves the unseen classes performance of the learned prompt comparing with CoOp, which indicates that penalizing loss sharpness for optimization is feasible for generalizable prompt learning. However, it is obvious that SCoOp suffers from a significant decline (3.39%) on the averaged seen classes performance. Although we carefully search the suitable perturbation radius to achieve the best performance for each single dataset, there is still a 0.87% performance gap between CoOp and SCoOp†. Therefore, directly applying SAM in prompt learning cannot alleviate the generalization trade-off problem.

To this end, we propose the GCSCoOp by constraining the optimizing gradient to have high relevance to both loss value and loss sharpness. Comparing with CoOp, GCSCoOp drastically improves the averaged unseen classes performance by 9.8% (even surpasses the results of Zero-shot CLIP) with only 0.22% degradation on seen classes, which **verifies that optimizing prompts with GCSAM can achieve our initial goal**, *i.e.*, improving the performance on unseen classes while maintaining the performance on seen classes.

Following with the prior arts, we also report the Harmonic Mean (HM) as the trade-off index. Comparing with the existing methods including the state-of-the-art KgCoOp, GCSCoOp achieves the highest HM among 10/11 datasets and also the highest averaged HM. GCSCoOp also significantly improves the HM performance against vanilla SCoOp by 2.13% in average. Even comparing with SCoOp†, GCSCoOp reaches a higher results on 6/11 datasets and surpasses the averaged results by 0.53%, while the results on other 5/11 datasets are also very close (most of the differences are less than 0.1%). Notably, HM index usually pays more attention on the unseen domain performance (the lower value) in calculation while GCSCoOp aims to maintain the seen domain performance. In other words, when GCSCoOp and SCoOp† obtain close results, GCSCoOp tends to have higher seen classes results with the cost of unseen classes decline in a lesser degree. Overall, the experi-

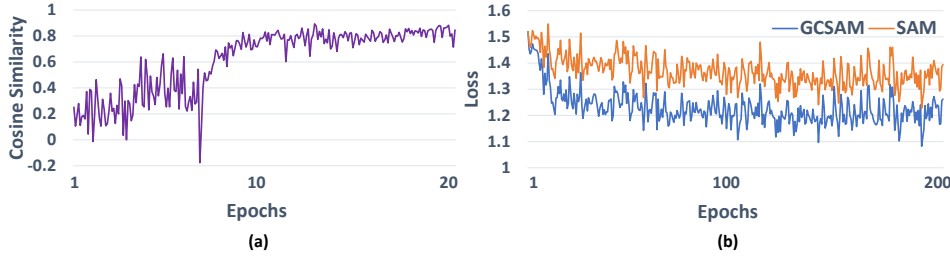

Figure 3: Training process visualization. Fig.3 (a) illustrates the cosine similarity between the gradient directions of $\mathcal{L}_{ERM}$ ($\mathcal{L}_{CE}$) and $\mathcal{L}_{SAM}$ during the training process on StanfordCars dataset. Fig.3 (b) demonstrates the loss value optimized by GCSAM and SAM on StanfordCars dataset.

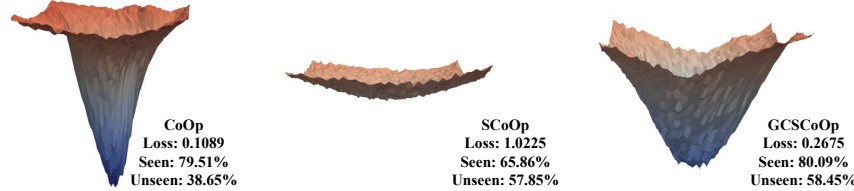

Figure 4: Loss landscape visualizations of CoOp, SCoOp and GCSCoOp on DTD dataset.

ments illustrated above verify that GCSCoOp can simultaneously improve the performance on both seen and unseen domains, and thus achieving better generalization trade-off in prompt learning task.

### 4.3 CROSS-DOMAIN GENERALIZATION

We also conduct the cross-domain generalization experiment. Specifically, we train the prompt on ImageNet and evaluate on four cross-domain datasets: ImageNetV2, ImageNet-Sketch, ImageNet-A and ImageNet-R. As demonstrated in Table.2, GCSCoOp achieves the best generalization performance on all of the four target domains comparing with existing approaches and SCoOp. Although GCSCoOp does not surpass the results on source domain against CoOp, the proposed method still improves both source and target performance simultaneously over SCoOp, indicating that adopting GCSAM in prompt learning can lead to a better generalization trade-off. These results on ImageNet dataset also introduce an intriguing observation. An appropriate and sufficient low loss value can not only enhance the discriminative ability of the learned prompt on source domain, but also on target domains, which verifies that loss value is a critical index in the trade-off problem from another perspective. We also provide more cross-dataset experimental results in Appendix A.4, which can further verify the effectiveness of GCSCoOp.

### 4.4 VISUALIZATION RESULTS

**Training process visualization**: We first compute the cosine similarity between the gradient direction of $\mathcal{L}_{ERM}$ and $\mathcal{L}_{SAM}$ during SCoOp's training process. As illustrated in Fig.3 (a), we witness that sometimes, especially in the early training stage, the directions of $G_{SAM}$ and $G_{ERM}$ have very limited consistency, or even conflict. This phenomenon causes a severe affect on loss value optimization, and provides the basic motivation support of our work to seek better optimization strategy that can optimize the loss sharpness and loss value simultaneously during the whole training process. As demonstrated in Fig.3 (b), GCSCoOp converges at a much lower loss value than vanilla SCoOp, thereby achieving better trade-off performance.

**Loss landscape visualization**: Following the instruction of Li et al. (2018), we plot the loss landscape visualization results of CoOp, SCoOp and GCSCoOp. In the main text, we only provide the clearest example which conducted on DTD dataset, and leave more comprehensive results in Appendix A.5. As illustrated in Fig.4, CoOp trained with traditional ERM easily generates a sharp loss landscape, which leads to the poor generalization ability of the learned prompt. However, if directly applying SAM in prompt learning, the optimizing gradient will have limited correlation or

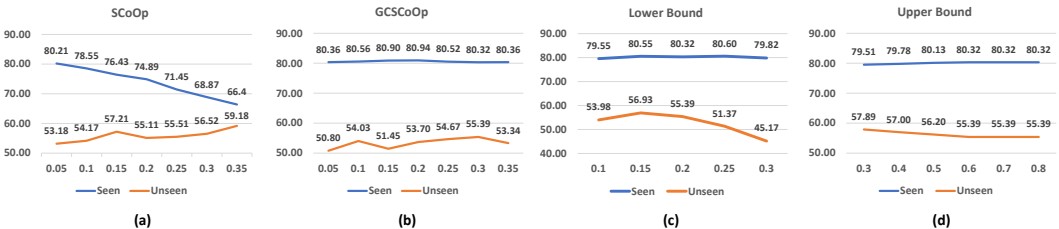

Figure 5: Fig.5 (a)&(b) illustrate the effect of perturbation radius $\rho$ on DTD dataset of SCoOp and GCSCoOp, respectively. Fig.5 (c)&(d) illustrate the effect of lower bound $\beta_1$ and upper bound $\beta_2$ on DTD dataset, respectively.

even conflict with the optimization objective of loss value. Therefore, although SCoOp can obtain a very flat loss landscape, but the minima is not able to converge to a sufficient low loss value, which severely affects the seen classes performance. By constraining the optimizing gradient to simultaneously have high relevance to the two-fold objective under different conditions, GCSCoOp obtains both flat loss landscape and low loss value at the same time, thereby achieving better generalization trade-off in prompt learning.

## 4.5 HYPERPARAMETER ANALYSIS

**Perturbation Radius:** As shown in Fig.5 (a), seen and unseen classes performance are in negative correlation when adjusting $\rho$ in SCoOp. This phenomenon indicates SCoOp can not achieve satisfying generalization trade-off since enlarging $\rho$ to improve unseen classes results will also cause the performance degradation on seen classes. However, as illustrated in Fig.5 (b), GCSCoOp can obtain very stable seen classes performance when adjusting $\rho$. Specifically, when moving $\rho$ from 0.05 to 0.35, GCSCoOp receives gradual improvements on unseen classes while maintaining the seen classes performance. However, when further adjusting $\rho$ from 0.3 to 0.35, we witness a slight decline on unseen classes results. Actually, this issue is caused by the lower bound threshold. Larger $\rho$ often generates smaller cosine similarity between $G_{SAM}$ and $G_{CE}$. If the $cos(\theta) \leq \beta_1$ condition often occurs during training, the optimization objective of loss sharpness tends to be neglected. To solve this problem, we can simply lower the value of $\beta_1$ or just choose a suitable $\rho$.

**Gradient Constrain Threshold:** As implemented in Appendix A.3.2, we choose $\beta_1 = 0.2$ and $\beta_2 = 0.8$ for GCSCoOp to learn prompt on DTD dataset. In this experiment, we adopt the control variates method to explore the effect of lower bound and upper bound, respectively. As shown in Fig.5 (c), at the beginning of raising $\beta_1$, both seen and unseen performances get improvements since loss value start to be considered during optimization. However, if continuously increase the value of $\beta_1$, loss value will be overemphasized in optimization, thus leading to a drastically decrease on unseen classes. $\beta_2$ is to constrain the effect of loss sharpness in optimization. As illustrated in Fig.5 (d), over-raising $\beta_2$ will gradually weaken the effect of loss sharpness, hence degrading the unseen classes performance. Generally, comparing Fig.5(b), (c) and (d) with (a), the sensitivity of the thresholds and perturbation radius in GCSAM are generally lower than the perturbation radius in SAM, which indicates that GCSCoOp is more stable with hyperparameters than vanilla SCoOp.

Comprehensive hyperparameter effect results in terms of *perturbation radius*, *gradient constrain thresholds*, *initialization*, *token-length* and *shot-length* are demonstrated in Appendix A.6.

## 5 CONCLUSION

This work targets an important problem in VLM prompt learning that tightly associates with real applications, *i.e.*, the generalization trade-off problem. We start from the optimization perspective to explore this problem, and conclude two indispensable factors for the trade-off problem: loss value and loss sharpness. By explicitly constraining the optimizing gradient to have high relevance to both loss value and loss sharpness during the whole optimization procedure, we finally propose the Gradient Constrained Sharpness-aware Context Optimization (GCSCoOp) for prompt learning, and effectively improve the generalization trade-off performance.

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

## A APPENDIX

The appendix of this paper is organized as follows. In Sec.A.1, we give in-depth analysis of what causes vanilla SAM fails to solve our proposed generalization trade-off problem in prompt learning, which also leads to the intuitive motivation of GCSCoOp. In Sec.A.2, we provide the detailed implementations of SCoOp and GCSCoOp within the pseudo codes. In Sec.A.3, we display the experimental setups including the dataset statistics and the hyperparameters of SCoOp and GC-SCoOp. In Sec.A.4, more cross-dataset generalization results are illustrated. In Sec.A.5, we provide more comprehensive loss landscape visualization comparison on multiple datasets. In Sec.A.6, we conduct various hyperparameter analysis experiments including perturbation radius, gradient constrain thresholds, shot-length, token-length and initialization. Finally in Sec.A.7, we summarize the limitations and future works of this paper.

### A.1 IN-DEPTH ANALYSIS OF SAM IN PROMPT LEARNING

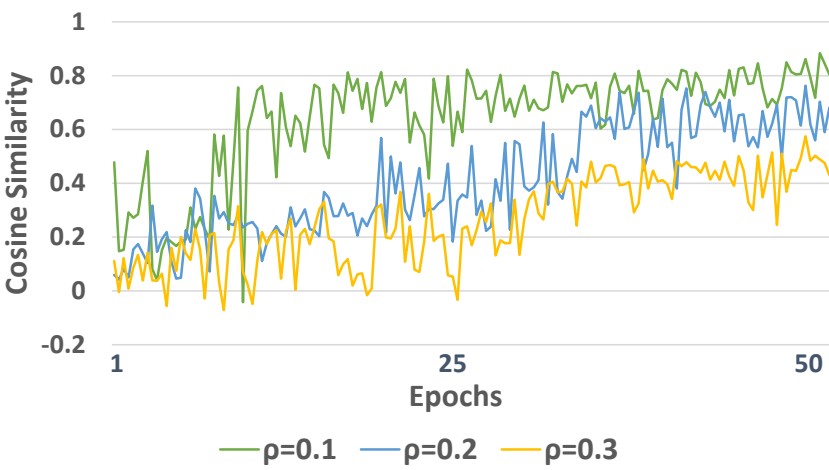

Figure 6: Cosine similarity between the gradient direction of $\mathcal{L}_{CE}$ and $\mathcal{L}_{SAM}$ of SCoOp under different perturbation radius $\rho$ on DTD dataset.

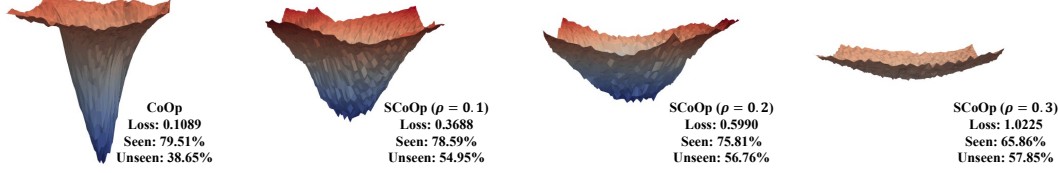

Figure 7: Loss landscape visualization of SCoOp under different perturbation radius $\rho$ on DTD dataset.

In this section, we give detailed explanations of *why SAM fails to solve the trade-off problem in prompt learning*. These analysis also lead to the intuitive motivation of GCSAM. To answer the above question, we first split it into two sub-questions according to previous discussions.

*Q1: Why does SAM fail to optimize loss sharpness and loss value simultaneously?*

To answer *Q1*, we explore the relationship between perturbation radius $\rho$ and the cosine similarity $\cos(\theta)$ (according to Eq.6). Obviously, larger $\rho$ penalizes a larger region of the neighborhood of minima to have uniform low loss, thus achieving flatter loss landscape and better generalization ability. However, as shown in Fig.6, we compare the cosine similarity $\cos(\theta)$ between the gradient direction of $\mathcal{L}_{CE}$ and $\mathcal{L}_{SAM}$ under $\rho = 0.1$, $\rho = 0.2$ and $\rho = 0.3$ conditions in SCoOp. We observe that larger $\rho$ leads to the smaller $\cos(\theta)$ during the whole optimization procedure. This observation indicates that achieving better generalization performance with flatter loss curve in SAM

will inevitably neglect the optimization objective of loss value. The above discussion directly reveals that loss sharpness and loss value cannot be appropriately and simultaneously optimized with the vanilla SAM in prompt learning.

*Q2: Why causes a much severe affect of Q1 in prompt learning than in overparameterized model training?*

SAM was initially designed to improve the generalization ability of overparameterized models. To answer *Q2*, we start from the characteristics of overparameterized model (Arora et al., 2019; Vicol et al., 2022). One of which is the underspecification problem, *i.e.*, given a learning problem with more parameters than datapoints, there are typically infinitely many solutions that achieve the optimal objective (Belkin, 2021). This problem reveals that a sufficient low loss value can be easily achieved when optimizing overparameterized models even under a relatively large $\rho$. However, there are very few tunable parameters in prompt leaning. In this paper, we adopt 4 token vectors while each token is 128-dim. Therefore, there are totally $4 \times 128 = 512$ parameters for optimization. Even the prompt learning is under few-shot learning, this task is still not able to meet with the underspecification condition, which means the loss value cannot converge to a sufficient low as easy as overparameterized models when using vanilla SAM for optimization. Concretely, as shown in Fig.8, the loss landscape for most of the datasets is single-peaked. The only special case is EuroSAT, which only has 80 training samples under 16-shot learning strategy and meets the underspecification condition.

Based on above discussion, we conclude that vanilla SAM is not able to solve the trade-off problem in prompt learning (verified by Fig.7), and thus we design the improved GCSAM to constrains the optimizing gradient to have high relevance to both loss value and loss sharpness for prompt learning task.

## A.2   PSEUDO CODES OF SCOOP AND GCSCOOP

Here, we provide the detailed training process of Sharpness-aware Context Optimization (SCoOp) and Gradient Constrained Sharpness-aware Context Optimization (GCSCoOp) in Algorithm.1.

## A.3   EXPERIMENTAL DETAILS

In this section, we give the detailed statistics of the evaluated datasets and the applied hyperparameter of SCoOp and GCSCoOp.

### A.3.1   DATASET STATISTICS

We describes the 15 datasets that has been exploited for evaluation. We report the task, number of classes, training and testing sample size in Table.3.

### A.3.2   HYPERPARAMETERS SELECTION FOR SCOOP AND GCSCOOP.

As demonstrated in Table.4, this section provides the hyperparameters including perturbation radius $\rho$, lower bound $\beta_1$ and upper bound $\beta_2$ thresholds that be applied in SCoOp, SCoOp† and GCSCoOp among different datasets. For GCSCoOp, as we mentioned before, we set $\rho = 0.1$, $\beta_1 = 0.5$ and $\beta_2 = 0.8$ as default, and these parameters are suitable with most of the datasets. However, there are still some datasets do not fit with the default parameters. To this end, we make small adjustments according to the characteristics of the datasets. Specifically, for dataset such as FGVCAircraft that cannot converge well on seen classes, we raise the thresholds to pay more attention on loss value during optimization. For datasets such as OxfordPets and Food101 that the gap between seen and unseen classes are not quite large (we witness the seen and unseen results are in positive correlation on these kind of datasets), we lower the thresholds to pay more attention on loss sharpness during optimization. For datasets such as DTD and EuroSAT that contain very expert information and hardly been seen when pre-training VLMs, we enlarge the perturbation radius and lower the lower bound to increase the generalizable ability on these small and expertise datasets. For fair comparison, we set the same perturbation radius $\rho$ for SCoOp and GCSCoOp. As for SCoOp†, we search $\rho$ from 0.005 to 2 to reach the best generalization performance on each dataset.

---

**Algorithm 1:** The training process of SCoOp and GCSCoOp

---

**Input:** Few-shot training set:$S \triangleq \cup_{i=1}^{n}\{(\boldsymbol{x}_i, \boldsymbol{y}_i)\}$, number of class categories $M$, class name
$\boldsymbol{c} = \{\boldsymbol{c}_j\}_{j=1}^{M}$, pre-trained CLIP model with image-encoder $\mathcal{I}$ and text encoder $\mathcal{T}$,
prompt token length $N$, training epoch $K$, learning rate $\eta$, perturbation radius $\rho$, lower
bound $\beta_1$ and upper bound $\beta_2$.

1 Initialize $\boldsymbol{v} = \{\boldsymbol{v}_i\}_{i=1}^{N}$;
2 **for** $k = 1, 2, , ..., K$ **do**
3  |  Generate text inputs based on $M$ class names with prompt: $\boldsymbol{t} = \{\{\boldsymbol{t}_j\}_{j=1}^{M}|\boldsymbol{t}_j = \{\boldsymbol{v}, \boldsymbol{c}_j\}\}$;
4  |  Obtain image features $\boldsymbol{f} = \mathcal{I}(\boldsymbol{x})$, text features $\boldsymbol{w} = \mathcal{T}(\boldsymbol{t})$;
5  |  Calculate the cross-entropy loss $\mathcal{L}_{CE}$ according to Eq.2 and obtain gradient $G_{CE}$;
6  |  Obtain the optimal perturbation $\hat{\boldsymbol{\epsilon}}$ according to Eq.4;
7  |  Calculate the local maximum loss $\mathcal{L}_{SAM}$ according to Eq.5 and obtain gradient $G_{SAM}$;
8  |  **if** *SCoOp* **then**
9  |  |  Update $\boldsymbol{v}^{k+1} = \boldsymbol{v}^k - \eta * G_{SAM}$;
10 |  **if** *GCSCoOp* **then**
11 |  |  Calculate the cosine distance $cos(\theta)$ between $G_{SAM}$ and $G_{CE}$ according to Eq.6;
12 |  |  **if** $cos(\theta) \leq \beta_1$ **then**
13 |  |  |  $G_{GCSAM} = G_{CE}$;
14 |  |  **else if** $\beta_1 < cos(\theta) < \beta_2$ **then**
15 |  |  |  $G_{\mathrm{mid}} = (G_{SAM} + G_{CE})/2$;
16 |  |  |  $G_{GCSAM} = \frac{G_{SAM} \cdot G_{\mathrm{mid}}}{\|G_{\mathrm{mid}}\|^2} G_{\mathrm{mid}}$;
17 |  |  **else if** $cos(\theta) \geq \beta_2$ **then**
18 |  |  |  $G_{GCSAM} = G_{SAM}$;
19 |  |  Update $\boldsymbol{v}^{k+1} = \boldsymbol{v}^k - \eta * G_{GCSAM}$;
20 **end**
**Output:** The parameters of prompt $\boldsymbol{v} = \{\boldsymbol{v}_i\}_{i=1}^{N}$.

---

Table 3: Detailed statistics of the evaluated datasets.

| Dataset | Task | Classes | Training Size | Testing Size |
|---|---|---|---|---|
| ImageNet | Object recognition | 1000 | 1.28M | 50000 |
| Caltech101 | Object recognition | 100 | 4128 | 2465 |
| OxfordPets | Fine-grained pets recognition | 37 | 2944 | 3669 |
| StanfordCars | Fine-grained car recognition | 196 | 6509 | 8041 |
| Flowers102 | Fine-grained flowers recognition | 102 | 4093 | 2463 |
| Food101 | Fine-grained food recognition | 101 | 50500 | 30300 |
| FGVCAircraft | Fine-grained aircraft recognition | 100 | 3334 | 3333 |
| SUN397 | Scene recognition | 397 | 15880 | 19850 |
| DTD | Texture recognition | 47 | 2820 | 1692 |
| EuroSAT | Satellite image recognition | 10 | 13500 | 8100 |
| UCF101 | Action recognition | 101 | 7639 | 3783 |
| ImageNet-V2 | Robustness of collocation | 1000 | N/A | 10000 |
| ImageNet-Sketch | Robustness of sketch domain | 1000 | N/A | 50889 |
| ImageNet-A | Robustness of adversarial attack | 200 | N/A | 7500 |
| ImageNet-R | Robustness of multi-domains | 200 | N/A | 30000 |

## A.4 CROSS-DATASET GENERALIZATION

As reported in Table.5, we learn the prompt on ImageNet and evaluate its generalization performance on other 10 datasets. Comparing with state-of-the-arts, SAM-based methods (SCoOp and GCSCoOp) achieve better averaged results. Moreover, GCSCoOp can make further improvements over SCoOp, which verifies the effectiveness of the proposed method.

Table 4: Hyperparameters applied in SCoOp, SCoOp† and GCSCoOp, respectively. Here, '-' indicates to remove the corresponding hyperparameter.

| | SCoOp | SCoOp† | GCSCoOp | | |
|---|---|---|---|---|---|
| | $\rho$ | $\rho$ | $\rho$ | $\beta_1$ | $\beta_2$ |
| ImageNet | 0.1 | 0.1 | 0.1 | 0.5 | 0.8 |
| Caltech101 | 0.1 | 2.0 | 0.1 | 0.5 | 0.8 |
| OxfordPets | 0.1 | 0.05 | 0.1 | - | 0.3 |
| StanfordCars | 0.1 | 0.03 | 0.1 | 0.5 | 0.8 |
| Flowers102 | 0.1 | 0.1 | 0.1 | 0.5 | 0.8 |
| Food101 | 0.1 | 0.2 | 0.1 | - | 0.3 |
| FGVCAircraft | 0.1 | 0.05 | 0.1 | 0.6 | 0.9 |
| SUN397 | 0.1 | 0.13 | 0.1 | 0.5 | 0.8 |
| DTD | 0.3 | 0.13 | 0.3 | 0.2 | 0.8 |
| EuroSAT | 0.2 | 0.1 | 0.2 | - | 0.8 |
| UCF101 | 0.1 | 0.08 | 0.1 | 0.5 | 0.8 |

Table 5: Performance comparison of VLM prompt learning methods on cross-dataset generalization.

| | CoOp | CoCoOp | ProGrad | KgCoOp | SCoOp | GCSCoOp |
|---|---|---|---|---|---|---|
| Caltech101 | 92.24 | 94.37 | 92.83 | 93.79 | 94.62 | 94.30 |
| OxfordPets | 89.12 | 90.45 | 89.49 | 89.78 | 90.07 | 90.64 |
| StanfordCars | 60.80 | 64.41 | 64.89 | 65.49 | 65.15 | 66.47 |
| Flowers102 | 67.59 | 71.44 | 69.65 | 69.96 | 71.01 | 70.91 |
| Food101 | 85.61 | 86.04 | 85.89 | 86.32 | 86.35 | 86.43 |
| FGVCAircraft | 15.33 | 20.95 | 18.49 | 22.68 | 21.02 | 22.82 |
| SUN397 | 62.91 | 66.44 | 64.44 | 66.42 | 67.18 | 67.37 |
| DTD | 40.88 | 44.72 | 43.54 | 46.20 | 44.13 | 44.52 |
| EuroSAT | 42.84 | 42.98 | 46.04 | 43.90 | 47.73 | 44.97 |
| UCF101 | 67.01 | 68.35 | 66.54 | 67.84 | 68.61 | 69.03 |
| AVG | 62.43 | 65.02 | 64.18 | 65.24 | 65.59 | 65.75 |

## A.5 SUPPLEMENTARY LOSS LANDSCAPE VISUALIZATION RESULTS

Fig.8 provides more comprehensive loss landscape visualization results of CoOp, SCoOp and GC-SCoOp on UCF101, Caltech101, StanfordCars, Flowers102 and EuroSAT datasets. In general, comparing with CoOp and SCoOp, the loss landscape generated by GCSCoOp are more capable to achieve both flat loss curve and sufficient low loss value simultaneously, which indicates that GCSCoOp successfully alleviates the problem of generalization trade-off in prompt learning.

## A.6 HYPERPARAMETER ANALYSIS

### A.6.1 PERTURBATION RADIUS

Fig.9 illustrates the performances of SCoOp and GCSCoOp under different perturbation radius $\rho$ on multiple datasets. The results demonstrate that GCSCoOp can effectively alleviate the generalization trade-off problem in SCoOp, and provide relatively stable seen classes performances when adjusting the perturbation radius. For unseen classes, when gradually enlarging the $\rho$, we witness that the performance generally rises first and then falls. This is because applying larger $\rho$ will lead to better generalization at the beginning. However, if continuously increase $\rho$, the cosine similarity $\cos(\theta)$ will decrease, thereby loss sharpness will be gradually neglected in optimization with the fixed gradient constrain thresholds.

### A.6.2 GRADIENT CONSTRAIN THRESHOLD

Fig.10 demonstrates the performances of GCSCoOp under different lower bound $\beta_1$ and upper bound $\beta_2$ on multiple datasets. The results verify that small $\beta_1$ and $\beta_2$ tend to focus on loss sharpness

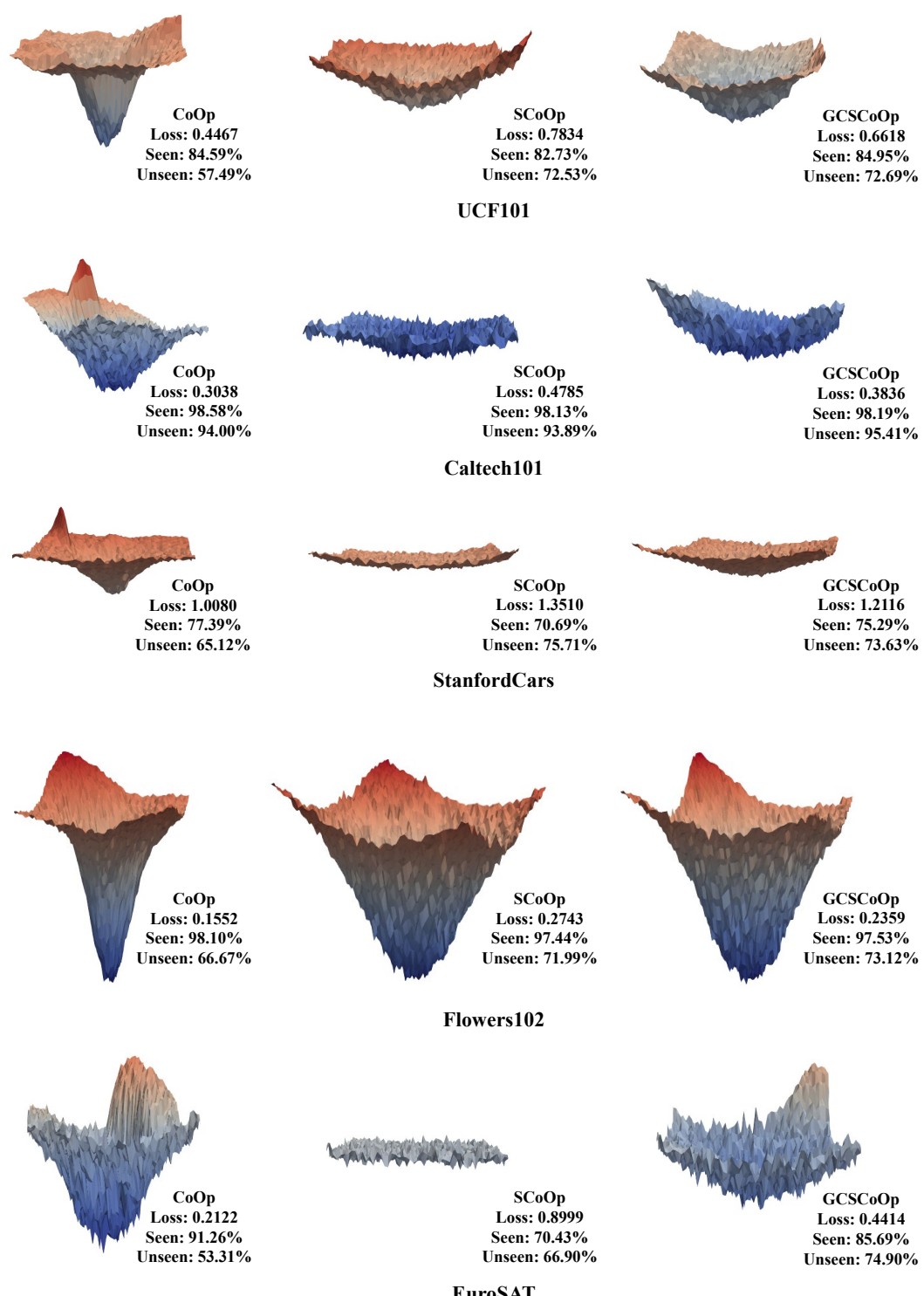

Figure 8: Comprehensive loss landscape visualization results of CoOp, SCoOp and GCSCoOp on UCF101, Caltech101, StanfordCars, Flowers102 and EuroSAT datasets, respectively.

during the optimization process, while large $\beta_1$ and $\beta_2$ tend to concentrate on loss value. Generally, the gradient constrain thresholds are in low sensitivity among different settings.

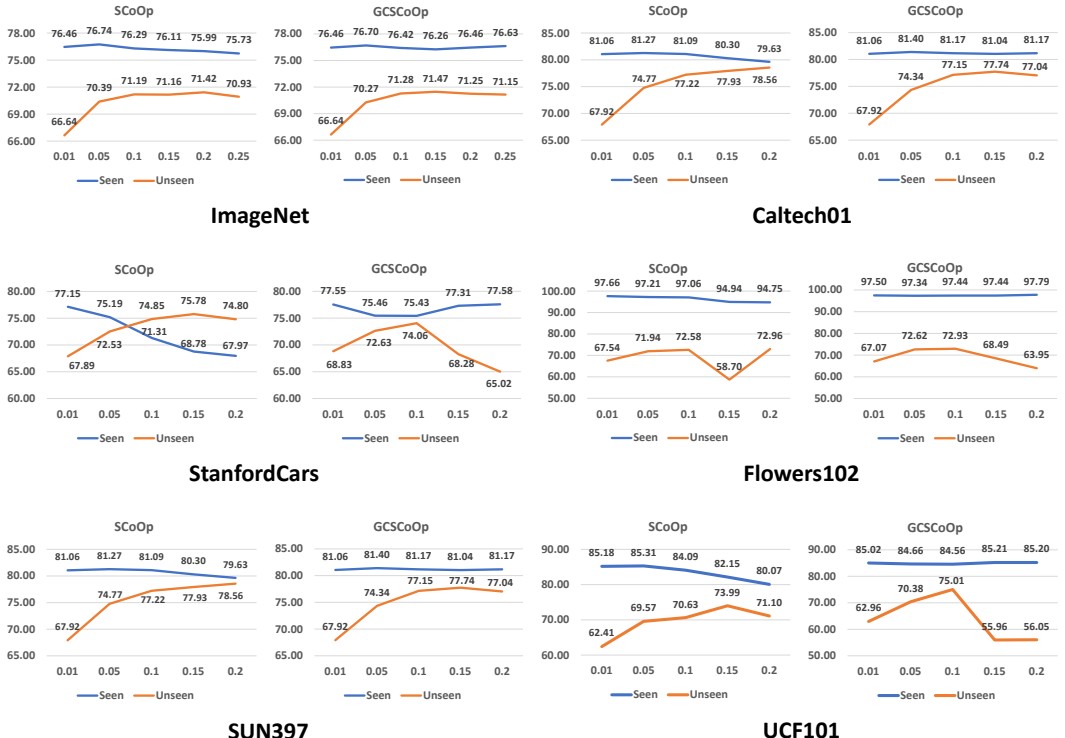

Figure 9: Perturbation radius $\rho$ effect analysis of SCoOp and GCSCoOp on ImageNet, Caltech101, StanfordCars, Flowers102, SUN397 and UCF101 datasets, respectively.

### A.6.3 SHOT-LENGTH AND TOKEN-LENGTH

Table.6 and Table.7 provide the performance of GCSCoOp under different shot-length and token-length, respectively. The shot-length represents the number of training samples per class that used in prompt learning. The token-length indicates the number of token vectors that applied as the learnable prompt. In the main experiments, we utilize 16 shots with 4 tokens as default.

As shown in Table.6, using less training samples will decrease the data diversity, thus inevitably undermining the prompt learning performance. Although the decline in most cases are slight and acceptable, we still witness that in very few datasets such as EuroSAT and UCF101, 4-shot learning severely decreases the performance on both seen and unseen classes. This problem may come from the unsuitable hyperparameters of perturbation radius or gradient constrain thresholds. We will explore this failure case in our future works.

Table.7 indicates that applying more tunable token vectors for prompt learning can slightly improve the seen classes performances, but also degrade the unseen classes performances. This phenomenon may come from the unsuitable hyperparameters of perturbation radius. More token vectors involve more tunable parameters. However, the perturbation radius is constrained by $||\epsilon||_p \leq \rho$, which relates to the total number of parameters. Therefore, using more tunable token vectors while just keeping the original $\rho$ will weaken the loss sharpness penalty on each of the single parameter.

### A.6.4 INITIALIZATION

Fig.11 demonstrates the performances of GCSCoOp with or without the initialization on multiple datasets. In default, we utilize "a photo of a [class name]" as the initialization template for prompt learning. When removing this initialization, we observe that seen classes performances slightly decrease while unseen classes slightly increase. These results indicate that initialization template is a useful regularization term to maintain the seen classes performance.

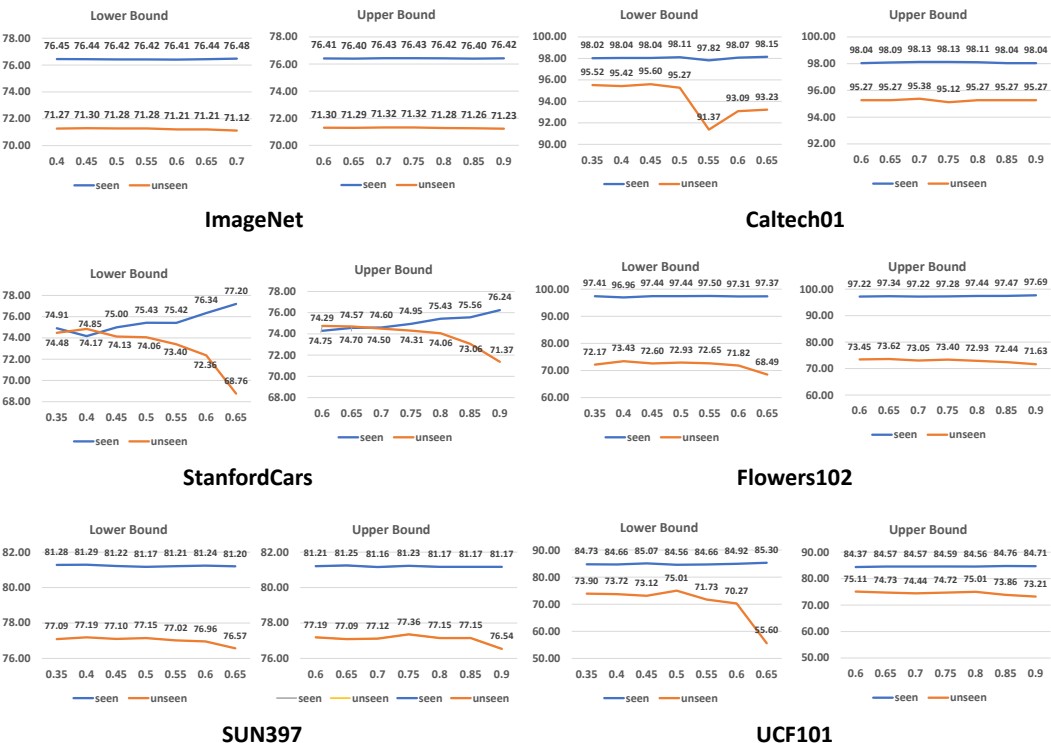

Figure 10: Gradient constrain thresholds $\beta_1$ & $\beta_2$ effect analysis of GCSCoOp on ImageNet, Caltech101, StanfordCars, Flowers102, SUN397 and UCF101 datasets, respectively.

Table 6: Shot-length effect analysis of GCSCoOp on 11 datasets.

| Dataset | Seen | | | Unseen | | |
|---|---|---|---|---|---|---|
| | 4-shot | 8-shot | 16shot | 4-shot | 8-shot | 16shot |
| ImageNet | 76.03 | 76.21 | 76.42 | 70.84 | 71.31 | 71.28 |
| Caltech101 | 97.55 | 97.57 | 98.11 | 93.74 | 93.38 | 95.27 |
| OxfordPets | 95.11 | 95.45 | 95.43 | 97.89 | 97.86 | 97.97 |
| StanfordCars | 69.49 | 74.79 | 75.43 | 69.64 | 73.06 | 74.06 |
| Flowers102 | 92.75 | 96.08 | 97.44 | 73.55 | 73.92 | 72.93 |
| Food101 | 90.66 | 90.70 | 90.96 | 91.69 | 91.82 | 92.07 |
| FGVCAircraft | 33.95 | 37.02 | 39.28 | 32.09 | 33.27 | 32.45 |
| SUN397 | 78.69 | 80.33 | 81.17 | 74.91 | 76.84 | 77.15 |
| DTD | 70.99 | 75.43 | 80.32 | 51.57 | 53.26 | 55.39 |
| EuroSAT | 78.55 | 79.37 | 87.91 | 48.59 | 58.60 | 69.92 |
| UCF101 | 79.35 | 82.69 | 84.56 | 69.57 | 69.28 | 75.01 |

## A.7 LIMITATIONS AND FUTURE WORKS

Although the default hyperparameters of perturbation radius and gradient constrain thresholds in GCSCoOp are effective with most of the downstream tasks (datasets), there are still some tasks need manual adjustments. Therefore, we will deeply explore the relationship and difference of the trade-off problem among different tasks according to the task characteristic, and consider how to adaptively set hyperparameters to make this approach more efficient and application-oriented in the future.

Table 7: Token-length effect analysis of GCSCoOp on 11 datasets.

| Dataset | Seen | | | Unseen | | |
|---|---|---|---|---|---|---|
| | N=4 | N=8 | N=16 | N=4 | N=8 | N=16 |
| ImageNet | 76.42 | 76.61 | 76.78 | 71.28 | 71.14 | 71.08 |
| Caltech101 | 98.11 | 98.06 | 98.08 | 95.27 | 95.52 | 94.65 |
| OxfordPets | 95.43 | 95.62 | 95.89 | 97.97 | 97.91 | 97.84 |
| StanfordCars | 75.43 | 72.96 | 76.61 | 74.06 | 74.27 | 72.67 |
| Flowers102 | 97.44 | 97.59 | 97.82 | 72.93 | 69.57 | 72.84 |
| Food101 | 90.96 | 90.93 | 90.79 | 92.07 | 91.88 | 92.06 |
| FGVCAircraft | 39.28 | 38.50 | 39.60 | 32.45 | 29.75 | 32.87 |
| SUN397 | 81.17 | 81.71 | 81.77 | 77.15 | 77.46 | 76.48 |
| DTD | 80.32 | 80.05 | 80.59 | 55.39 | 55.80 | 51.65 |
| EuroSAT | 87.91 | 86.46 | 89.34 | 69.92 | 62.81 | 60.47 |
| UCF101 | 84.56 | 84.82 | 84.33 | 75.01 | 69.03 | 73.50 |

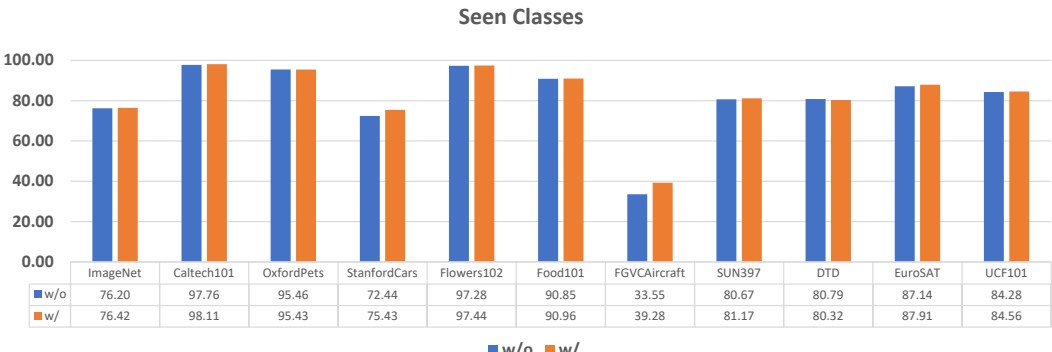

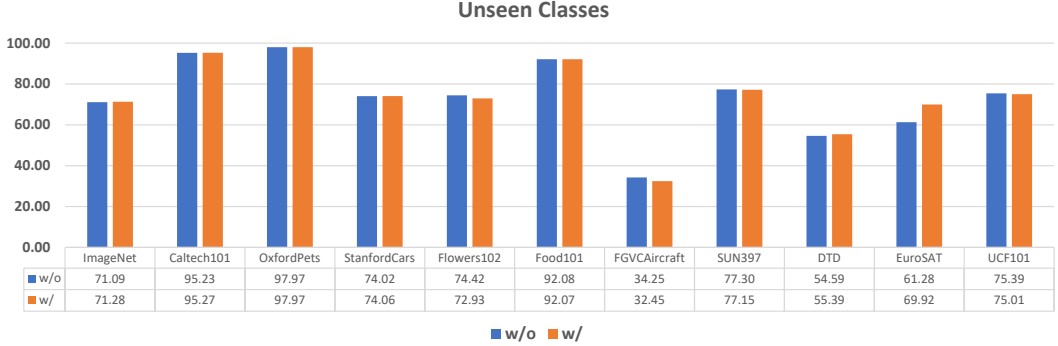

Figure 11: Performance comparison between **w/** and **w/o** initialization of GCSCoOp on 11 datasets.

