# OpenReview forum: "Gradient Constrained Sharpness-aware Prompt Learning for Vision-Language Models"
_ICLR.cc/2024/Conference — ICLR 2024 Conference Withdrawn Submission_

### Official Review · Reviewer_DgeK · 2023-10-25

**Soundness:** 3 good
**Presentation:** 4 excellent
**Contribution:** 2 fair
**Rating:** 5
**Confidence:** 4

**Summary:**

The authors propose to a method to adaptively interpolate between the SAM gradient and the ERM gradient. The ERM gradient is taken by itself if the two gradients disagree two much (as measured by the cosine similarity). The authors observe that large disagreements usually occur early during training. The authors argue that their method finds deeper minima than SAM but flatter minima than vanilla ERM. This strikes a nice tradeoff between generalization and risk minimization.

**Strengths:**

- Very easy to understand.
- Writing is good; well organized.
- Method is intuitive. Figure 3 and 4 show that the authors' method is a balance between sharpness minimization and loss minimization. I found these figures helpful.

**Weaknesses:**

- The contributions are overstated. e.g. in the first sentence of the abstract, the authors state that "existing generalizable methods neglect seen classes degradation". Throughout the paper the authors emphasize that they redefine the problem as a tradeoff problem where they claim to maintain performance on seen classes while improving performance on unseen classes.
  + The tradeoff problem between seen and unseen class accuracy is very well explored by prior work in this area, starting with CoOp and CoCoOp. All of these methods have to deal with this trade-off between novel and base class accuracies. This is why all the prior work report harmonic means. As you train longer, the base class accuracy continues to increase while the novel class accuracy degrades. This is why the bolded number for "seen" and "unseen" classes usually occur for different methods in Table 1. This trade-off problem is certainly not new.
- The method is simple and not hard to think of. It is similar in spirit to [SAGM].
- The results seem impressive at first glance (Table 1), but I have some serious concerns regarding their integrity. How hard did you tune CoOp? The authors of KgCoOp reported a average HM of 74.6 on CoOp in their Table 3, this is compared to 71.4 in the proposed paper. Why is there a 3% degradation? What happens when you run SCoOp with rho=0.0, does it agree with your CoOp numbers? I'm not sure how much performance gain is coming from improvements in the method versus how much performance gain is coming from the proposed method being tuned harder.
- There is a sentence towards the end of the introduction that says: "Different from overparameterized models that theoretically have multiple optimal solutions, learnable prompts have very few tunable parameters, i.e., a set of token vectors in text inputs, which
means the optimal solution in prompt learning task is very limited. Therefore, the loss value is much
more sensitive with regard to the direction of optimizing gradient in prompt learning." I disagree with this and find it very misleading. Do you have evidence to back it up? There are many good solutions to the prompt learning problem in different parts of the parameter space; you could verify that just by testing different handcrafted prompts.

[SAGM] Pengfei Wang et al. "Sharpness-Aware Gradient Matching for Domain Generalization"

**Questions:**

None.

---

### Official Review · Reviewer_GqHE · 2023-11-01

**Soundness:** 2 fair
**Presentation:** 3 good
**Contribution:** 2 fair
**Rating:** 5
**Confidence:** 3

**Summary:**

This paper proposed a novel sharpness-aware based optimization method to improve the performance of vision-language models. More specifically, this paper tried to enhance the performance of VLM on unseen classes while maintaining the performance on seen classes. The experimental results illustrate that the proposed method can obtain a better performance on most seen-to-unseen generalization  tasks.

**Strengths:**

1. This paper focuses on an important problem about maintaining the performance of VLM on seen tasks.
2. Sharpness-Aware Minimization is an important method to improve the generalization of neural network and this paper further investigate the application of SAM on VLM generalization.
3. The proposed method is very easy to follow.

**Weaknesses:**

1. The proposed method tries to achieve a trade-off between $G_{SAM}$ and $G_{CE}$.  I think the key idea is very similar to the method [1], which tries to use $\alpha G_{SAM} + (1-\alpha) G_{CE}$ to update parameters. I think the method in [1] can also solve the trade-off problem: "only if loss value and loss sharpness can be optimized appropriately and simultaneously, the learned prompt can achieve satisfying trade-off." Maybe you should provide more explanation about that and compare the performance with [1].

2. There are many papers trying to further improve the performance of SAM and maybe they can also achieve a great performance on these VLM generalization tasks, such as GSAM [2]. Maybe you should provide more results about that.

3. For the experimental results, I think the improvement is not very significant for me. It is very closed in some tasks and I'm not sure whether 3 random seeds is enough.


[1] Penalizing Gradient Norm for Efficiently Improving Generalization in Deep Learning. ICML 2022

[2] Surrogate gap minimization improves sharpness-aware training. ICLR 2022

**Questions:**

1. The motivation is not very clear to me. I understand you can use vanilla SAM to improve the generalization. However,  this claim is not very convincing for me: "the directions of ∇LSAM and ∇LERM have very limited consistency (a large direction angle), or even conflict (the angle is larger than 90◦)." I think $∇L_{SAM}$ contains two parts: the first is about $∇L_{ERM}$ and the second is $G_{flat}$, which help the model converge to a flat region. So I think these two gradients don't need to be very consistent since the main difference is $G_{flat}$, especially when the model converges to a sharp local minima and we need to focus more on $G_{flat}$ to escape from the local minima and not $G_{CE}$.

---

### Official Review · Reviewer_SWWN · 2023-11-01

**Soundness:** 2 fair
**Presentation:** 2 fair
**Contribution:** 2 fair
**Rating:** 5
**Confidence:** 4

**Summary:**

In this paper, the authors proposed GC-SCoOp, a Gradient Constrained Sharpness-aware Context Optimization approach for prompt learning. Extensive experimental results and visualizations showed improved performance over multiple baseline methods.

**Strengths:**

1. The idea of optimizing both loss value and loss sharpness for prompt learning makes sense and the proposed GC-SCoOp is also technically sound to me.
2. The authors conducted multiple experiments on several datasets against a few baseline methods to show the effectiveness of the proposed method. Loss landscape visualizations also verified that the proposed approach indeed optimize both the loss value and loss sharpness.

**Weaknesses:**

1. The comparison result in Table 2 is mixed with a large portion settings that the proposed GCSCoOp is actually worse than other baseline methods.
2. The proposed method heavily depends on hyper-parameter ρ, β1 and β2 and it seems that performance is not quite stable with respect to these parameters.
3. Symbols are not consistent. For example, in Figure 2(b) the authors used v_t but v_k was used in section 3.2.
4. Writing needs to be improved as there are several typos. For example, 'Benefited from the rich nature language supervision' on page 3 should be 'Benefited from the rich natural language supervision'.

**Questions:**

1.  Why the results are mixed in Table 2?
2.  What would be a good strategy for hyper-parameter setting?

---

### Official Review · Reviewer_wX3D · 2023-11-01

**Soundness:** 3 good
**Presentation:** 2 fair
**Contribution:** 3 good
**Rating:** 5
**Confidence:** 5

**Summary:**

This paper targets at generalizable prompt learning for vision-language models and leverage Gradient Constrained Sharpness-aware Minimization to balance the loss sharpness and the loss value for better generalization performance.

**Strengths:**

1. The idea of using loss sharpness along with loss value to guide the text prompt learning is interesting.
2. Several experiments are conducted and support the proposed approach well.

**Weaknesses:**

1. The gradient modification approach is similar to Prograd, where gradient projection[1] is a widely used trick in continual learning. I wonder why we need two hyper-parameters to split the gradient direction into three segments.
If we just follow[1] and use the projection of $G_{SAM}$ on $G_{CE}$ when the two gradients conflict, will the result be better or worse?
2. I am sorry I can't admit that KgCoOp is the state-of-art approach. CoPrompt[2], PLOT[3], MaPLe[4], VioLET[5], and CoCoOpter[6] all outperform KgCoOp on 11 average visual recognition tasks. I think maybe you mean that KgCoOp is the sota among the single text modal approaches. However, this is not clearly claimed in the paper.
3. As the results of multi-modal prompt approaches, such as CoPrompt[2], MaPLe[4], and VioLET[5] outperform your proposed approach significantly, it seems that multi-modal prompt is vital for CoOp based approaches. I wonder whether your proposed GCSAM can improve the multi-modal prompt approaches if your proposed approach is orthogonal to multi-modal prompt approaches.
4. The proposed approach aims to improve the performance on unseen classes while maintaining the performance on seen classes.
Then can this approach improve few-shot performance similar to CoOp and PLOT[3]?
5. Presentation could be improved. For example, Figure 2 serves as the centrepiece of the paper's methodology, but the legend lacks explanatory notes, which detracts from the paper's readability.
[1] Yu T, Kumar S, Gupta A, et al. Gradient surgery for multi-task learning[J]. Advances in Neural Information Processing Systems, 2020, 33: 5824-5836.
[2] Roy, Shuvendu, and Ali Etemad. "Consistency-guided Prompt Learning for Vision-Language Models." arXiv preprint arXiv:2306.01195 (2023).
[3] Chen G, Yao W, Song X, et al. Prompt learning with optimal transport for vision-language models[J]. arXiv preprint arXiv:2210.01253, 2022.
[4] Khattak, Muhammad Uzair, et al. "Maple: Multi-modal prompt learning." Proceedings of the IEEE/CVF Conference on Computer Vision and Pattern Recognition. 2023.
[5] Wang Y, Liu Y, Zhang X, et al. VioLET: Vision-Language Efficient Tuning with Collaborative Multi-modal Gradients[C]//Proceedings of the 31st ACM International Conference on Multimedia. 2023: 4595-4605.
[6] Yan J, Xie Y, Guo Y, et al. CoCoOpter: Pre-train, prompt, and fine-tune the vision-language model for few-shot image classification[J]. International Journal of Multimedia Information Retrieval, 2023, 12(2): 27.

**Questions:**

Please refer to the weaknesses. If the author can address my question well and I am glad to raise my score.